# Central amygdala circuitry modulates nociceptive processing through differential hierarchical interaction with affective network dynamics

Isabel Wank[1,4], Pinelopi Pliota[2,4], Sylvia Badurek [3], Klaus Kraitsy[3], Joanna Kaczanowska [2], Johannes Griessner[2], Silke Kreitz[1], Andreas Hess[1,5] & Wulf Haubensak [2,5 ✉]

The central amygdala (CE) emerges as a critical node for affective processing. However, how CE local circuitry interacts with brain wide affective states is yet uncharted. Using basic nociception as proxy, we find that gene expression suggests diverging roles of the two major CE neuronal populations, protein kinase C δ-expressing (PKCδ[+]) and somatostatin-expressing (SST[+]) cells. Optogenetic (o)fMRI demonstrates that PKCδ[+]/SST[+] circuits engage specific separable functional subnetworks to modulate global brain dynamics by a differential bottom-up vs. top-down hierarchical mesoscale mechanism. This diverging modulation impacts on nocifensive behavior and may underly CE control of affective processing.

[1] Institute of Experimental and Clinical Pharmacology and Toxicology, Friedrich-Alexander University Erlangen-Nuremberg, Erlangen, Germany. [2] Research Institute of Molecular Pathology (IMP), Vienna Biocenter (VBC), Vienna, Austria. [3] Preclinical Phenotyping Facility, Vienna Biocenter Core Facilities GmbH (VBCF), Vienna, Austria. [4] These authors contributed equally: Isabel Wank, Pinelopi Pliota. [5] These authors jointly supervised this work: Andreas Hess, Wulf Haubensak. ✉email: wulf.haubensak@imp.ac.at

The central amygdala (CE) is a canonical relay in processing aversive signals, most notably related to fear and anxiety but also pain. It is functionally divided into central lateral (CEl) and central medial (CEm), with CEl receiving nociceptive input from the parabrachial nucleus, thalamus, and cortex, and CEm as output, regulating defensive behavior and nociception[1–3]. CEl is composed of at least two antagonistic neuronal populations marked by expression of (i) protein kinase C δ (PKCδ), enkephalin, and oxytocin receptors and (ii) somatostatin (SST), dynorphin, and corticotropin-releasing hormone. This antagonistic circuit has been extensively studied in fear and anxiety[4]. Its role in nociception has only been addressed very recently, with SST[+] neurons[5] and (populations largely overlapping with)[6] PKCδ[+] neurons[5] as direct modulators of nocifensive behavior. Thus CEl SST[+]/PKCδ[+] circuitry emerges as a central element for both fear and nociception[3], two highly related basic survival processes. However, despite dense mapping of CE circuit connectivity and function[7], the mesoscale mechanisms by which these local circuitries interact with global brain dynamics remain uncharted. Such mesoscale understanding fills a critical gap, linking CE circuit level mechanisms to global functional brain states. To explore this, we focused on nociceptive processing, the most basic (as compared to more complex affective states) entry point into this problem.

Gene expression profiles of PKCδ[+] and SST[+] CEl populations showed the differential distribution of nociception-associated genes, suggesting that these two CEl neuronal subtypes have diverging roles in pain processing. Using ofMRI, we demonstrated that PKCδ[+]- and SST[+]-driven CEl circuits differentially modulate perceptive and behavioral responses to noxious stimuli through bottom-up or top-down interaction with brain-wide functional connectivity. Concluding, those findings were backed up by nociception-related behavioral tests using chemogenetic activation of the two CEl neuronal subtypes using DREADDs (designer receptors exclusively activated by designer drugs).

Summed up, our findings from genetics, modulated brain circuits as well as behavioral data support differential mesoscale mechanisms and most likely opposing roles for the two most prominent CEl neuronal populations, PKCδ[+] and SST[+] neurons, in regulating aversive brain states.

## Results and discussion

To explore potential functions of PKCδ[+] and SST[+] CEl populations in the context of pain, we first screened deep sequencing data from FACS-sorted CEl PKCδ[+] or SST[+] cells obtained from naive adult male C57BL6/J mice[8] for the differential expression (DE) of known nociception-related genes (Supplementary Data 3, DE Pain gene set). Among these, 54 genes were expressed in any of these populations (transcript per million: TPM > 1), with 7 genes specifically expressed in PKCδ[+], and 14 in SST[+] cells (Fig. 1a). Between the two populations, 21 genes showed significant differential expression (Supplementary Fig. 1a). Considering the molecular interaction network of these genes, CEl PKCδ[+] and SST[+] neurons showed diverging distribution along molecular signaling pathways (Fig. 1b). Within this molecular pain network, SST[+] neurons expressed key transmitters of pain (e.g., pain-related sodium channel voltage-gated type VIII alpha subunit (*Scn8a*)[9,10] and dynorphin (*Pdyn*)[11], at the center nodes of this network), whereas PKCδ[+] cells support more modulatory aspects of this network (adenosine A3 receptor (*Adora3*)[12], protein kinase A (PKA)-RIIβ[13], peripheral network nodes). Ingenuity Pathway Analysis (IPA) of the DE of these 54 expressed genes identified nociception as a key disease function (as expected) and further revealed differential effects of PKCδ[+] cells vs. SST[+] neurons on nociception (Supplementary Fig. 1b). Taken together,

expression, and differential distribution, of nociceptive genes, supported that these two CEl neuronal subtypes are involved in nociception with potentially diverging roles.

To understand how these local circuits modulate global brain-wide nociceptive processing in vivo, we combined cell-type-specific optogenetic manipulation at defined timepoints with whole-brain stimulus-driven BOLD (blood oxygenation level-dependent) fMRI (ofMRI). To this end, we injected PKCδ::Cre and SST::Cre mice stereotactically with Cre-dependent viruses expressing either GFP (green fluorescent protein) for controls or ChR2 (channelrhodopsin 2) into the right CEl for optogenetic activation (Supplementary Figs. 2a and 3). OfMRI stimulation consisted of three conditions: heat-only (general nociception), laser-only (identification of brain regions engaged directly by CEl), and simultaneous laser-heat application (co-stimulation; modulation of nociception by CEl) (Supplementary Fig. 2a).

We first assessed if and how CEl neurons modulate global nociceptive processing by comparing classical BOLD amplitudes (Fig. 2a) between CEl PKCδ[+]/SST[+] activation and respective GFP controls for the whole brain, and for specific brain regions known to play a role in amygdala-driven behavioral circuits as well as nociception[14] (Fig. 2b). The amplitude maps did not show major differences in signal distribution between heat-only and laser-heat co-stimulation (Fig. 2a). Compared to the rather strong heat-only, the weaker laser-only effects suggest a primarily modulatory role of CE activation, in line with CE inhibitory and peptidergic signaling[15]. As expected, between-group contrasts filtered out a clear overall interaction of the co-stimulation of the ChR2 groups to the respective GFP controls. The activation patterns of both GFP controls were similar and differed notably from ChR2 groups, where PKCδ[+] showed the least activation of all groups, indicating an overall antagonistic interaction of PKCδ[+] with nociceptive brain states. Co-stimulation of CEl PKCδ[+] (Fig. 2b, gray dotted lines with triangles) reduced heat-evoked BOLD amplitudes in the brainstem, thalamus, the limbic system (including the amygdala) and the basal forebrain (septum, diagonal band of Broca, nucleus accumbens, pallidum), indicating that the amygdala (via PKCδ[+]) downregulated the ascending nociceptive input directly (via thalamus and basal forebrain), and inhibited its progress to the higher-order regions by an antinociceptive bottom-up modulatory approach. Notably, the (primary) somatosensory response was only slightly enhanced compared to GFP controls, implying that the lateral nociceptive system, which mainly locates incoming nociceptive stimuli, remained mostly unchanged. Opposingly, CEl SST[+] co-stimulation (Fig. 2b, black solid lines with circles) showed no effect on the input from the brainstem, thalamus, or the basal forebrain, but evoked enhanced BOLD amplitudes to heat in the somatosensory cortex and the limbic system, indicating that higher-order regions were more active than in GFP controls. The enhanced amplitude in somatosensory cortices may indicate that the stimuli were perceived as more noxious, compared to GFP controls.

As it is difficult to draw conclusions about modulatory implications of changed static BOLD patterns alone, we investigated how PKCδ[+]/SST[+] activation changed the dynamic interaction within brain regions based on functional connectivity (FC, see "Methods") tested against respective GFP controls. First, we assessed the direct brain-wide modulatory effects of the two CEl populations by optogenetic activation alone (Fig. 3a, b, left and Supplementary Fig. 4a), identifying the brain regions directly addressed by the respective CEl activation, and next the modulatory effect of those regions on cerebral nociceptive processing (Fig. 3a, b, right and Supplementary Fig. 4b), by combining laser application with simultaneous noxious thermal stimulation at the hind paw. Compared to the respective GFP controls, laser application (Fig. 3a, b, left and Supplementary Fig. 4a) of CEl

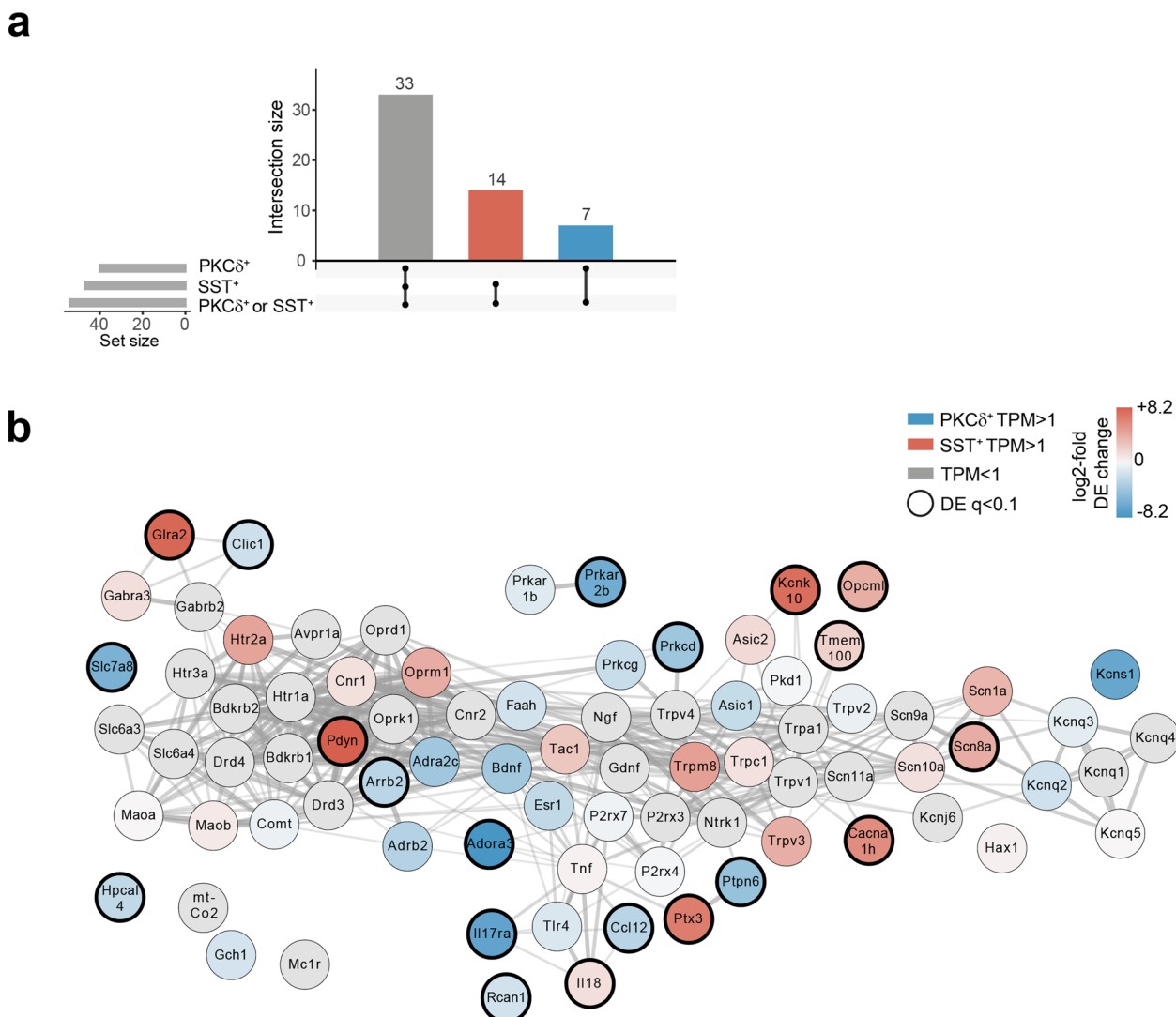

**Fig. 1 Expression of pain-related genes in CEl PKCδ⁺ and SST⁺ neurons. a** Expression of pain-related genes expressed in CEl. Distribution of 54 expressed (transcripts per million, TPM > 1) genes (Supplementary Data 3) in PKCδ⁺ (blue) or SST⁺ populations (red). **b** CEl PKCδ⁺ or SST⁺ neurons differentially express (DE) components of the molecular interaction network from the pain-related gene set (Supplementary Data 3). Blue or red node color represents DESeq2 log2FoldChange of the expressed genes (**a**) in either of the cell types, thick node border marking genes with significant differential expression with adjusted P value padj < 0.1. Genes in the network not expressed (TPM < 1) are gray. For abbreviations of genes, see Supplementary Data 3.

PKCδ⁺ significantly reduced FC within the brainstem, thalamus, and basal forebrain. CEl SST⁺ on the other hand also led to an overall reduction of FC compared to controls but had its major effects on higher-level cortical (sensory, association, cingulate, motor) and limbic regions (amygdala, hypothalamus, and basal forebrain) without affecting the brainstem.

Those regions, directly addressed by laser application, interfered with central processing of noxious heat differentially for both neuronal populations (Fig. 3a, b, right and Supplementary Fig. 4b): compared to their GFP controls, CEl PKCδ⁺-co-stimulation significantly attenuated FC of mainly lower-level regions like basal forebrain (to/from thalamus, cortex, hippocampus, brainstem, hypothalamus) and hypothalamus (to/from thalamus, cortex, hippocampus, amygdala, brainstem). As the cortex was not directly influenced by laser application alone, this pattern is likely driven via the lower-level structures thalamus, basal forebrain, and brainstem. Looking at the co-stimulation networks, cortical FC to/from the thalamus, hypothalamus, and basal forebrain, but not brainstem, was reduced. Taking directed structural connectivity[16] into account, this suggested that cortical

activity was modulated bottom-up via the thalamus, hypothalamus, and the basal forebrain, with the two latter ones receiving direct inhibitory synaptic input from PKCδ⁺ CEl neurons. Opposingly, SST⁺ co-stimulation reduced mainly FC of the sensory cortex (to/from thalamus, cingulum, hippocampus, amygdala) and brainstem (to/from the hippocampus, motor cortex), modulating nociception in a top-down fashion via cortex. Interestingly, the main effect of both populations was a net decrease of FC compared to GFP controls, potentially reflecting the inhibitory action of those GABAergic neuronal populations.

Comparing the heat-processing network of wild-type (wt) mice (Supplementary Fig. 5a) with those connections modulated by CEl PKCδ⁺/SST⁺-co-stimulation (Fig. 3b), 12 out of 38 connections overlapped with CEl PKCδ⁺, but only three connections overlapped with CEl SST⁺. Direct comparison (Supplementary Fig. 5b) of CEl PKCδ⁺/SST⁺ laser + heat 50 °C co-stimulation with wt 45 °C or 50 °C revealed that CEl PKCδ⁺ co-stimulation showed stronger FC compared to wt 45 °C, but weaker FC than wt 50 °C. This indicates that CEl PKCδ⁺ activation diminishes the perception of noxious heat. CEl SST⁺ co-stimulation, on the

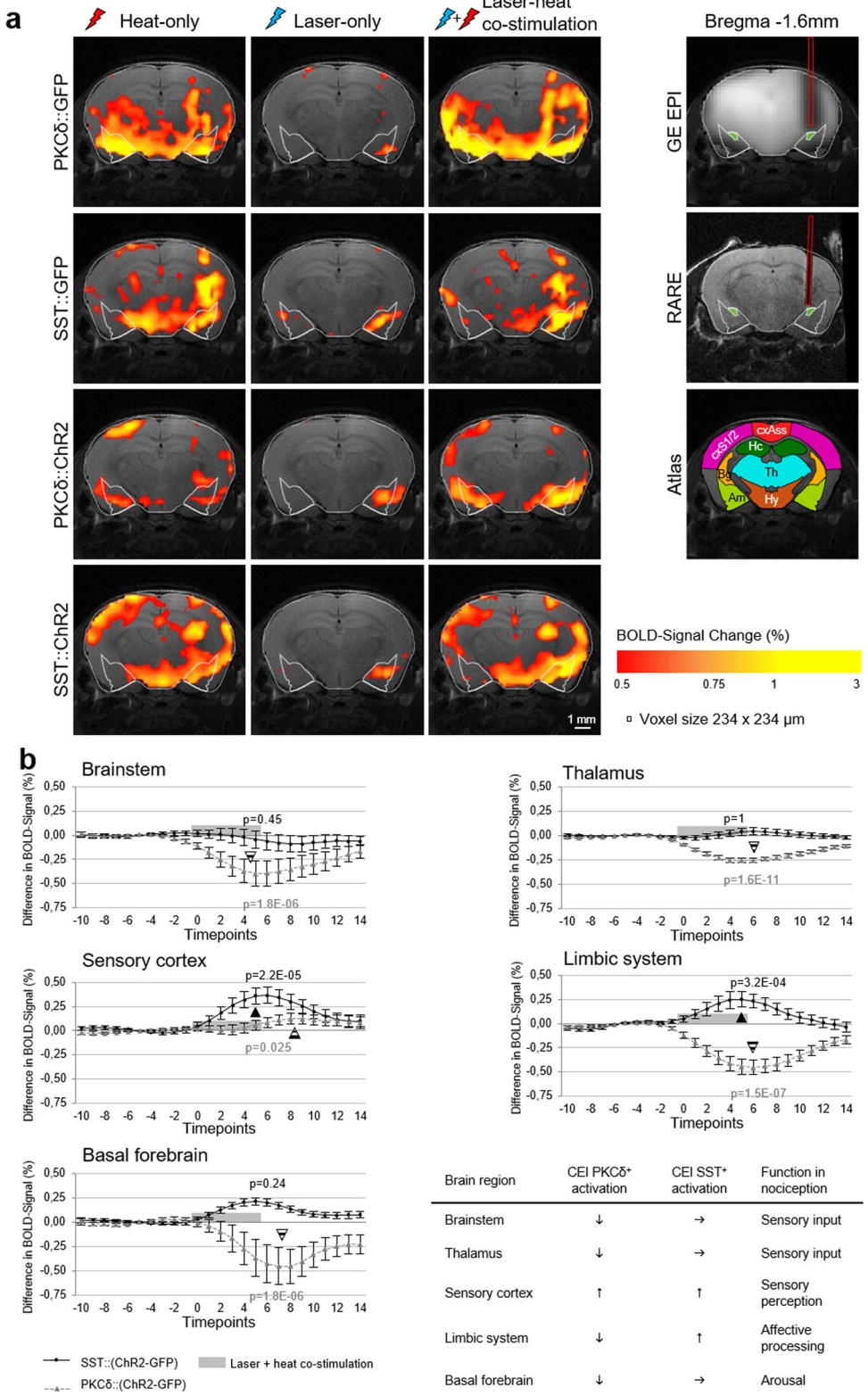

other hand, displayed much stronger FC than wt 45 °C, and even stronger FC than wt 50 °C. These findings support the notion that CEl PKCδ⁺ and SST⁺ neuronal bottom-up vs. top-down interaction with global brain states has net antagonistic effects on nociception.

Indeed, DREADD chemogenetic activation of the respective cell types (PKCδ::M3/ SST::M3; see "Methods", Supplementary Fig. 6) differentially modulated nociception-related behavior.

Mice received either saline or clozapine-N-oxide (CNO) prior to performing the hot plate test or von Frey test (Supplementary Fig. 6b, c). After saline injection, the average filament force evoking withdrawal did not differ between groups. However, after CNO injections, the required force for SST::M3 mice to react was significantly different from PKCδ::M3 mice, while the required force for PKCδ::M3 mice was significantly greater than PKCδ:: GFP or SST::GFP controls. We obtained a similar trend for the

**Fig. 2 BOLD signal response amplitudes. a** BOLD signal response amplitudes for all three stimulation conditions. Activation patterns for both GFP-control groups were comparable and differed from the pattern of the ChR2 groups ($n_{PKC\delta::GFP} = 5$, $n_{SST::GFP} = 3$, $n_{PKC\delta::ChR2} = 9$, $n_{SST::ChR2} = 6$). Note that the GFP controls during laser-only conditions may reflect low heat and/or light-related responses inherent to this technology[35]. As these are consistent across groups and treatment contrasts, this should not interfere with the interpretation of results. Heat-induced artifacts were previously shown to exhibit a characteristic logarithmic time course[36]. We analyzed the time courses of the right amygdala for all four groups (data not shown) and found only the typical hemodynamic response-like signal phenotype with comparable kinetics and amplitudes. Apparently, differences between heat-only and co-stimulation were marginal in all groups. PKCδ+ showed the most spatially confined signal distribution. Depicted in the most right column are examples of gradient-echo EPI and anatomical RARE slides including the optical fiber (original size shown as a red box), which appears enlarged in EPI due to susceptibility artifacts. Also shown is a modified compact version of the Paxinos[31] mouse brain atlas to help with the interpretation of the data. Exemplary high-resolution RARE anatomy of a standard reference mouse was used as background for all signal maps. Overlayed is a reduced atlas mask with the amygdala outlined and the CEl marked in green. **b** Shown are differences in average BOLD response amplitudes evoked by laser-heat co-stimulation (time window marked as a light gray box) between ChR2 groups and the corresponding GFP controls ($n_{PKC\delta::GFP} = 5$, $n_{SST::GFP} = 3$, $n_{PKC\delta::ChR2} = 9$, $n_{SST::ChR2} = 6$). Gray dotted line with triangles: laser-heat co-stimulation of CEl PKCδ+ neurons significantly reduced (▼) BOLD response amplitudes evoked by peripheral heat in all observed subnetworks compared to the respective GFP controls, with exception of the sensory cortex, where amplitudes were similar in both groups (the seemingly delayed rise in ChR2 amplitude is due to differential kinetics when returning to baseline after stimulation, with ChR2 returning more slowly). Black solid lines with circles: In contrast, activation of SST+ neurons enhanced (▲) the BOLD response amplitudes of the sensory cortex and limbic system, but had no effect on the brainstem and thalamus. Summary table: when compared to the respective GFP controls, activation of CEl PKCδ+ reduced incoming nociceptive input from the periphery and as consequence nociceptive processing in higher-order brain regions in a bottom-up modulatory effect. Activation of CEl SST+ had an opposing effect, as the input remained unchanged but perception and nociceptive processing were enhanced compared to controls, in line with top-down modulation of noxious heat processing (see also Supplementary Table 1). Statistical significance was verified by corrected one-factor repeated-measures ANOVA (see "Methods"); *P* values are noted next to the corresponding time series. Error bars display SEM for a difference of means across animals and brain regions. Am amygdala, Bg basal ganglia, cxAss association cortex, cxS1/2 primary/secondary somatosensory cortex, Hc hippocampus, Hy hypothalamus, Th thalamus.

hot plate test, where PKCδ::M3 mice performed a jump reaction at a higher temperature when comparing between groups under similar conditions after CNO injection (Supplementary Fig. 6d, e). We note that an overall increase in temperature tolerance observed in the GFP groups reflects habituation to this assay upon repeated exposure and/or CNO effects. This notwithstanding, the PKCδ+ and SST+ manipulations showed diverging trends also in this measure. This was further reflected by a larger fraction of SST::M3 than GFP and PKCδ::M3 mice displaying a jump reaction during the CNO treatment (Supplementary Fig. 6f). Overall, these results are in line with the idea of opposing effects of PKCδ+ and SST+ cells in pain, as described recently in a set of somewhat incongruent studies. One study described SST+ neurons as anti- and PKCδ+ as pro-nociceptive[5], while another publication found a fraction of cells largely overlapping with PKCδ+[6] to be antinociceptive. Our findings align better with Hua et al.[6], who found that a population of cells largely overlapping with PKCδ+ neurons in CEl enhanced withdrawal thresholds to mechanical stimulation, noxious heat, and noxious cold. The diverging behavioral results suggest that CEl may gate nociceptive behavior in a state-dependent manner, depending on the experimental variation and as consequence different behavioral states of the animal in the respective experiments. Taken together, CEl PKCδ+ vs. SST+ microcircuitry modulates global affective brain states with antagonistic effects in nociceptive behavior, through differential bottom-up vs. top-down mesoscale interactions.

In summary, we identified CEl as a key relay that expresses genes linked to the modulation of nociception. These gene sets were in part differentially expressed in the two major CEl PKCδ+ and SST+ neuronal subpopulations, suggesting functionally divergent roles for the two CEl populations in nociception. Most of the functional impact of nociception-linked genes are related to intercellular signaling (Supplementary Fig. 1b). Due to this differential expression and intracellular signaling makeup, these cell types differentially sample and process ascending input (e.g., from periaqueductal gray (PAG)) to CE. A straightforward interpretation would be that the molecular makeup promotes neuronal activation in SST+ cells, whereas it attenuates activation of PKCδ+ neurons, in response to (the same) incoming nociceptive

signals. In turn, this increases the inhibition of SST+ onto PKCδ+ neurons, potentially disinhibiting CE output to the brainstem, while suppressing PKCδ+-mediated bottom-up modulation of cortical pain states via the basal forebrain[17].

To investigate how these canonical interactions coupled into global nociceptive brain states, we used a combination of optogenetics, chemogenetics, and fMRI. We found that selective activation of those populations indeed modulated nociceptive brain states and nocifensive behavior differentially, engaging divergent brain regions and mesoscale mechanisms. fMRI functional connectivity analysis helped to gain insight into those complex, poorly understood mesoscale networks and revealed hierarchical features. Overall, both cell types exerted a predominantly negative effect on functional connectivity, for both laser-only and laser-heat co-stimulation. This might reflect that these two inhibitory populations suppressed the activity of their direct synaptic targets thereby modulating functional connectivity. CEl PKCδ+ neurons modulated nociception in a bottom-up fashion via the thalamus and the basal forebrain, producing a widespread decrease of functional connectivity in striatal-cortical loops (basal forebrain, cortex, thalamus, hippocampus, and hypothalamus), controlling thereby cortical activity. However, CEl SST+ activation primarily targeted limbic-cortical connections. The network modulated by SST+ neurons appeared more precisely defined and consisted of the amygdala, cortical regions (sensory-motor and cingulate cortex), hippocampus, and brainstem; regions involved in the cognitive-affective aspects of nociception. It is well known that attention, expectation, previous experience, and emotional state of the subject have a great influence on the perception of pain mediated by cortical and limbic brain regions and their descending pathways[18]. This top-down modulation of nociception/pain works in both directions and can promote analgesia[19,20] as well as proalgesia[21,22].

Strikingly, these differential mesoscale interactions may reflect the local asymmetry between PKCδ+/SST+ cells in CE circuitry. We propose that, in analogy to aversive fear signals, nociceptive information may activate SST+ cells[4,23–25]. This activation facilitates aversive responses by disinhibiting CEm outputs to PAG via local inhibition of PKCδ+ neurons and/or direct interaction with PAG and brainstem. This nociceptive information flow is

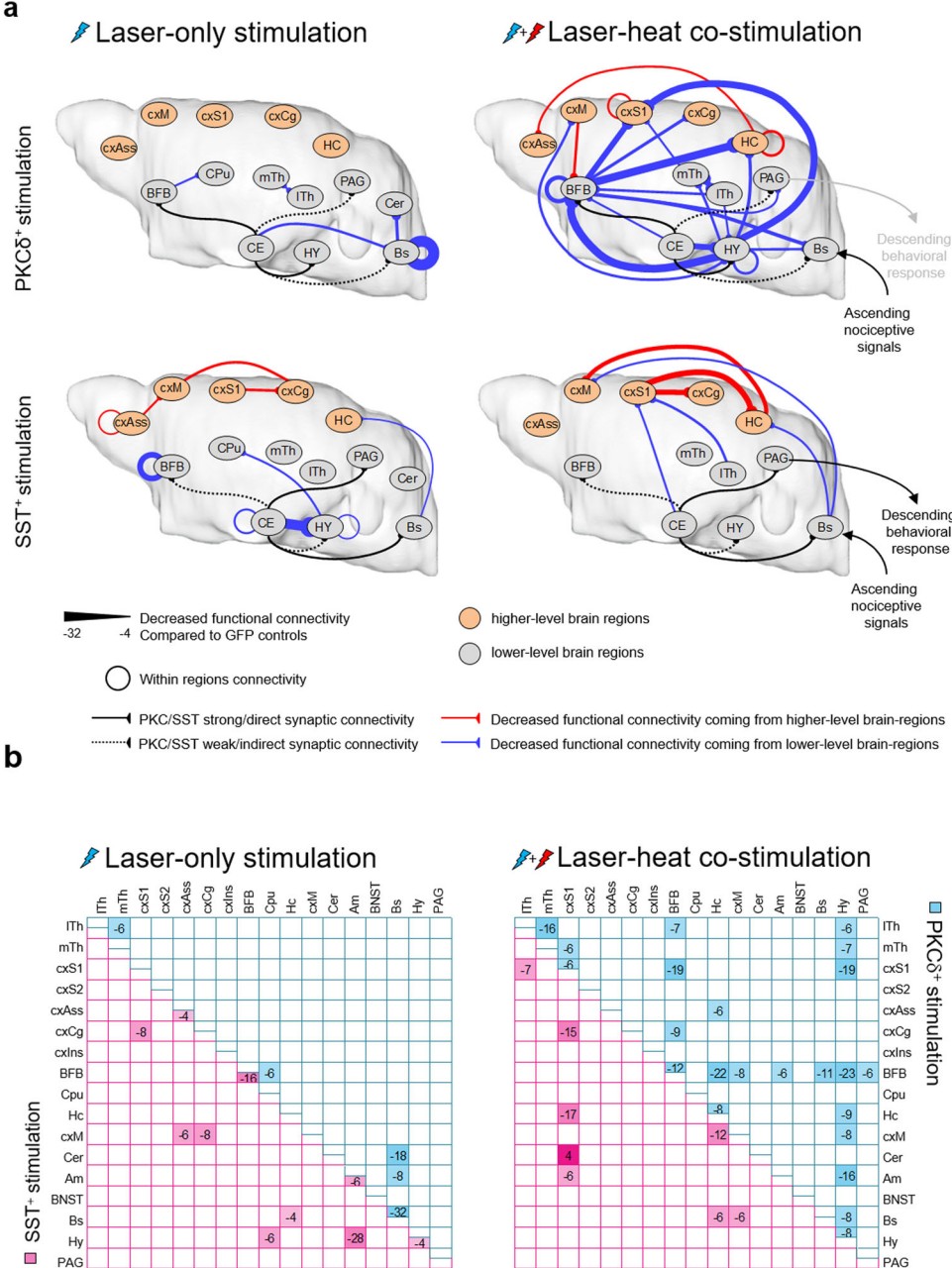

antagonized by PKCδ+ activity, which inhibits CEm outputs and suppresses aversive states in cortical networks bottom-up by uncoupling primary sensory and cingulate cortex from subcortical nociceptive states via direct synaptic projections to the basal forebrain. We note that PKCδ+ projections to the basal forebrain gate the affective value of environmental stimuli in cortical areas[17]. Thus, PKCδ+ activity may antagonize aversive brain states (pain) by uncoupling the cortical affective experience from the subcortical primary sensory component of pain. As mentioned above, these PKCδ+ actions, in turn, are suppressed by pain signals in SST+ cells and their local inhibition of PKCδ+ activity in CE.

In a general context, this study delineates a molecular-to-systems level framework that identifies hierarchically distinct bottom-up and top-down interaction as the mesoscale mechanism by which CEl modulates nociceptive processing (Supplementary Table 1), the most basic form of affective processing. We, therefore, propose the divergent bottom-up and top-down

hierarchical network interactions observed here as a universal motif by which CEl populations differentially gate competing for affective brain functions[26], from pro-/anti-nociception to fear[4]/reward[27] and active/passive[28,29] behavioral decisions. Lastly, our study illustrates how local, neighboring neuronal populations engage differential mesoscale interactions with dynamic brain-wide states, not evident from the pure anatomical hierarchy of CE circuits. Such mesoscale mechanisms provide a critical link between circuit neuroscience and systems-wide brain states, also for translational research (fMRI in psychology and psychiatry).

## Methods

**Animal description and housing**. Eight to twelve-week-old C57BL/6J or transgenic male mice (PKCδ::GluClα-CRE BAC transgenic mice (PKCδ::Cre) or Srt-ires-CRE (SST::Cre) (Jackson Laboratory stock no: 013044) backcrossed to C57BL/6J) were used as indicated (Tables 1 and 2). The animals were weaned at day 21–23 after birth and food and water were provided ad libitum, while they were housed in groups of a maximum of five animals per cage at 21 °C in a 14-h light/10-h dark cycle-dependent of daylight savings time. Tests were performed during the light

**Fig. 3 Summarized changes in functional connectivity (FC) evoked by optogenetic or laser-heat co-stimulation in CEl (compared to respective GFP controls). a** Pure optogenetic activation of CEl PKCδ$^+$ neurons (left top) reduced FC within the lower-level (light gray spheres) brain regions thalamus, basal forebrain, and brainstem. Optogenetic activation of CEl SST$^+$ neurons (left bottom) reduced FC in three distinct subnetworks: one higher-level (orange spheres) cortical (cxAss, cxCg, cxM, cxS1) and two lower-level subcortical (Am and Hy; CPu, Hc, and Bs). Optogenetic modulation of nociception using laser-heat co-stimulation of PKCδ$^+$ neurons (right top) modulated a vast network of brain regions, particularly a circuit involving the amygdala, hypothalamus, hippocampus, basal forebrain, and primary sensory cortex. Even though FC is generally undirected, possible directions may be derived from the underlying directed structural connectivity[16]. This indicated that CE controlled BFB and Hy as key nodes, which then modulated the higher-level brain regions. Co-stimulation of SST$^+$ neurons (right bottom) during heat stimulation influenced a very confined network of regions. Structural connectivity indicated that CE controlled higher-level cortical regions (cxS1, cxM, cxCg) and Hc directly and indirectly via Bs. Projection paths of CEl PKCδ$^+$ and SST$^+$ neurons indicated by solid black lines = major and direct, dotted black lines = minor and indirect. To ease interpretation of the data, changes in FC were pseudo-directed ("synapses" forming on the target region), using information from anterograde tracing studies (Allen brain atlas[16]). The unit of FC used here represents summed up significant changes between CEl PKCδ$^+$, SST$^+$ and their corresponding GFP controls in negative and positive correlations (homoscedastic two-tailed Student's $t$ test, $P \leq 0.05$, uncorrected; compared to the matching controls; $n_{PKCδ::GFP} = 5$, $n_{SST::GFP} = 3$, $n_{PKCδ::ChR2} = 9$, $n_{SST::ChR2} = 6$). Meaning, a net FC of −6 here in this picture represents six significantly different connections (positive and/or negative Pearson r) that were greater in GFP than in the ChR2 group, or, for example, seven that were greater and one that was smaller in GFP. Shown: changes stronger than 4 (laser) or 6 (nociception) connections for a clear picture. Two regions showed increased FC (cxS1 with Cer). As all other changes resulted in reduced connectivity, this connection was omitted in this picture. Regions not involved in any network are not shown. **b** Corresponding connectivity matrices for Fig. 3a. Sum of changes (homoscedastic two-tailed Student's $t$ test, $P \leq 0.05$, uncorrected) in negative (−1) as well as positive (+1) correlations for laser application (left) and laser-heat co-stimulation (right). Shown are changes stronger than 4 (laser) or 6 (nociception) connections in blue for PKCδ$^+$ and pink for SST$^+$ activation compared to the respective GFP controls. Am amygdala, BFB basal forebrain including septum, diagonal band of Broca, nucleus accumbens, pallidum, BNST bed nucleus of stria terminalis, Bs brainstem, CE central amygdala, reflected by the respective neuronal populations, PKCδ$^+$ or SST$^+$, Cer cerebellum, CPu caudate putamen, cxAss parts of association cortex, cxCg cingulate cortex, cxIns insular cortex, cxM motor cortex, cxS1 prim. somatosensory cortex, cxS2 secondary somatosensory cortex, Hc hippocampus, Hy hypothalamus, lTh lateral thalamus, mTh medial thalamus, PAG periaqueductal gray.

**Table 1 Group assignments and sample sizes for behavior combined with DREADD chemogenetic activation.**

| Group name | Mouse strain | Stereotactically injected construct and titer (GC/ml) | Virus source | Number (+excluded) |
|---|---|---|---|---|
| PKCδ::GFP | PKCδ::CRE Tg(Prkcd-glc-1/CFP,-cre)EH124Gsat MGI:3844446[4] | AAV2/5.EF1a.DIO.GFP.WPRE (2.1 × 10$^{12}$) | Vienna Research Institute of Molecular Pathology (IMP) | 13 (+1) |
| PKCδ::M3 | PKCδ::CRE Tg(Prkcd-glc-1/CFP,-cre)EH124Gsat MGI:3844446[4] | AAV2/5.hSyn.DIO.hM3.mCherry.WPRE.hGh (2.1 × 10$^{12}$) | University of Pennsylvania | 15 (+1) |
| SST::GFP | SST::CRE Jackson Stock No:028864 Sst-IRES-Cre knock-in (C57BL/6J)[37] | AAV2/5.EF1a.DIO.GFP.WPRE (2.1 × 10$^{12}$) | Vienna Research Institute of Molecular Pathology (IMP) | 7 (+1) |
| SST::M3 | SST::CRE Jackson Stock No:028864 Sst-IRES-Cre knock-in (C57BL/6J)[37] | AAV2/5.hSyn.DIO.hM3.mCherry.WPRE.hGh (2.1 × 10$^{12}$) | University of Pennsylvania | 7 (+1) |

**Table 2 Group assignments and sample sizes for ofMRI.**

| Group name | Mouse strain | Stereotactically injected construct and titer (GC/ml) | Virus source | Number (+excluded) |
|---|---|---|---|---|
| PKCδ::GFP | PKCδ::CRE Tg(Prkcd-glc-1/CFP,-cre) EH124Gsat MGI:3844446[4] | AAV2/5.EF1a.DIO.GFP.WPRE (2.1 × 10$^{12}$) | Vienna Research Institute of Molecular Pathology (IMP) | 5 |
| PKCδ::ChR2 | PKCδ::CRE Tg(Prkcd-glc-1/CFP,-cre) EH124Gsat MGI:3844446[4] | AAV2/5.hsyn.hChR2(H134R).eYFP.WPRE (1.2 × 10$^{13}$) | University of Pennsylvania | 9 (+1) |
| SST::GFP | SST::CRE Jackson Stock No:028864 Sst-IRES-Cre knock-in (C57BL/6J)[37] | AAV2/5.EF1a.DIO.GFP.WPRE (2.1 × 10$^{12}$) | Vienna Research Institute of Molecular Pathology (IMP) | 3 (+2) |
| SST::ChR2 | SST::CRE Jackson Stock No:028864 Sst-IRES-Cre knock-in (C57BL/6J)[37] | AAV2/5.hsyn.hChR2(H134R).eYFP.WPRE (1.2 × 10$^{13}$) | University of Pennsylvania | 6 (+4) |

cycle. All animal procedures were performed in accordance with institutional guidelines and were approved by the respective Austrian and German authorities covered by the license M58/002220/2011/9. In this study, due to breeding schemes and colony design, only male mice were used. We note that the results may not be applicable to both sexes, which has to be repeated by testing female mice under the same conditions.

**Group assignment and sample sizes**

*Behavior.* To activate CEl PKCδ$^+$/SST$^+$ cell populations in vivo during behavioral tests, the less disturbing DREADD approach was chosen instead of optogenetics. DREADDs can be activated by intraperitoneal injection of Clozapine-N-oxide (CNO, see below section Behavioral Tests). Animals from each genotype were randomly assigned to either control- or M3 (expressing DREADDs under control of the human synapsin promoter) group (Table 1). For the behavioral experiments,

30 PKCδ::Cre and 16 SST::Cre were injected with M3- or green fluorescent protein (GFP)-expressing virus (Table 1). As responsiveness and variance were comparable between both GFP groups, animals were pooled to enhance statistical power (Supplementary Fig. 6g–k).

Specifically, four animals with incorrect viral expression were excluded from the analysis after histological examination, where the experimenter was not blinded to the assignment of the groups, as viral expression between test- and control group is different. Successful viral expression was assessed using PKCδ staining for histological control, as PKCδ is expressed in CEl (in the amygdala) and not in neighboring areas, so when the viral expression was inside the limits of PKCδ staining area then the injection was considered successful, otherwise the mouse was excluded from the analysis (Supplementary Fig. 3).

*ofMRI.* In total, 15 PKCδ::Cre and 15 SST::Cre were injected with GFP- or ChR2-expressing virus (Table 2). Of the 15 animals per mouse strain, five animals were

randomly assigned to the control group (injected with cre-dependent AAV expressing GFP) and 10 to the ChR2 group (injected with cre-dependent AAV expressing ChR2). The group size of the fMRI experiments was designed in a 1:2 ratio, with twice the number of ChR2-expressing animals. The rationale for this decision was that we designed the experiment for analysis with pooled control groups to gain statistical power while reducing animal numbers. Retrospectively, pooling of control groups was not necessary for fMRI. To keep with the overall cohort and criteria, we kept all measured animals instead of matching sample sizes post hoc. In total, seven animals were excluded from analysis due to multiple reasons: five animals lost the implanted optogenetic fiber, one animal died before measurement and one had a malformed hind paw that prevented correct and secure fixation of the Peltier heating element.

**Viral targeting, fiber implantation, and histology.** Mice 8–12-week-old were deeply anesthetized with isoflurane (5%, Isoflo, Abbot Laboratories). Surgeries were performed under stereotaxic control (Model 1900 with Equipment, David Kopf Instruments) and anesthetics (1.6–2%) were constantly supplied through the nosepiece throughout the duration of the surgery. Body temperature was kept constant to 36 °C by a heating pad controlled by a rectal thermometer. For viral injections, first subcutaneous anesthesia was applied locally (Xylanaest 1%, Gebro Pharma), an incision was made to reveal the skull, which was then drilled with a stereotaxic mounted drill (Model 1911, David Kopf Instruments). For virus injections, the Micro4 Syringe Pump Controller (World Precision Instruments) was used, and the injection rate was 10 nl/min to a total volume of 100 nl. For behavior, viruses were bilaterally injected using a glass capillary Nanoliter 2000 injector (World Precision Instruments) at the following coordinates: CEl anterior–posterior −1.35 (from bregma), medial–lateral 2.75 (from midline), dorsal–ventral 4.7 mm (Table 1).

For fMRI, manipulations were targeted to the right amygdala (Table 2). In total, 200–400-μm ferrule-connected optogenetic fiber stubs (Doric Lenses, MFC200 or MFC400) were implanted 0.3 μm above the injection site of the right CEl and stabilized with SuperBond dental cement (SuperBond C&B kit with L-Polymer, Prestige Dental Products Ltd). Individual fiber positions upon CEl were confirmed based on anatomical MRI (Supplementary Fig. 2c). Directly after surgery, mice were left for recovery until they were fully awake and then put back to their home cage. After surgery, the drinking water was mixed with Carprofen (Rimadyl, Pfizer, 250 mg/l) and Enrofloxacin (Baytril, KVP pharma, 400 mg/ml) for 10–14 days. At least 4 weeks were allowed for viral expression and recovery until the start of the experiments.

For histology, after behavioral experiments were completed, mice were anesthetized with intraperitoneally Ketamine/Xylazine (OGRIS Pharma Vertriebs-GmbH, 10 mg/ml in PBS, OGRIS Pharma Vertriebs-GmbH, 1 mg/ml) until reflexes were completely absent. Then they were transcardially perfused with heparin solution (Sigma-Aldrich, 10 U/ml Heparin/PBS) and 4% PFA in PBS. The brains were extracted and submerged in 15% sucrose in PBS (Sucrose crystals, Fluka Biochemika) overnight. The next day, the brains were covered in a cryoprotective embedding medium (Tissue Tek, Sakura Finitech B.V.) and were frozen using dry ice. The frozen brains were sliced in 20-μm coronal sections using a cryostat. For staining, slides were initially permeabilized in PBST (PBS plus 0.1% Triton X-100, Sigma-Aldrich) for 10 min. Unspecific binding was blocked using BSA for 30 min. PKCδ staining was performed overnight at 4 °C (primary antibody: anti-PKCδ (IgG 2b), 610398, BD Biosciences, 1:1000 in BSA). The next day, after three washes with PBST, the secondary antibody and DAPI were applied for 2 h (anti-mouse IgG, A-21052, Life technologies 1:1000 in BSA; DAPI, D3571, Invitrogen, 1:1000 in BSA) followed by three PBST washes. Slides were covered with Fluorescence Mounting Medium (Dako, S302380) and coverslipped for imaging.

**Neuronal population sequencing data.** Reads from previously published neuronal population sequencing data[8] (GEO: GSE95154) were trimmed using trimgalore v0.5.0, and reads mapping to abundant sequences included in the iGenomes UCSC GRCm38 reference (mouse rDNA, mouse mitochondrial chromosome, phiX174 genome, adapter) were removed using bowtie2 v2.3.4.1 alignment. The remaining reads were aligned to the mouse genome (Ensembl GRCm38 release 94) using star v2.6.0c, and reads in genes were counted with featureCounts (subread v1.6.2). Genes with more than one average transcript per million (TPM) were considered to be expressed in the population.

These genes were further analyzed for association with nociception and/or presence in a known pain gene list[30] (Fig. 1a) and analyzed for differential gene expression using raw counts and DESeq2 v1.26.0 (Supplementary Fig. 1). These data were used to build a molecular interaction network for pain-related genes (Fig. 1b). Specifically, a STRING network of the 84 pain genes (Supplementary Data 3, DE) was generated using stringApp 1.6.0 in Cytoscape 3.8.1.

These differentially expressed genes were further analyzed using Ingenuity Pathway Analysis (IPA)(Qiagen)(Supplementary Fig. 1b, IPA). This analysis identified "nociception" (among others) as a significantly associated term in "Disease or Functions Annotation" (Supplementary Data 3).

**Behavioral testing**

*von Frey test.* Tests were performed on a Model 37450 electronic dynamic plantar aesthesiometer (Ugo Basile, Varese, Italy) with a 0.5-mm diameter increasing force filament. Animals were habituated to the measuring apparatus for 2 h before behavioral testing. Saline or Clozapine-N-oxide (CNO (Sigma)) was administered intraperitoneally (i.p.) at 10 mg/kg in saline during the habituation phase, 30 min prior to testing. Animals were tested with 0–10 g increasing force at a rate of 0.5 g/s. The force needed to evoke a twitching or lifting response was measured. Values were averaged over three trials of each left and right hind paw.

*Hot plate test.* Tests were performed on an electronically controlled hot plate analgesia apparatus (IITC Life Science, Woodland Hills, USA). Animals were injected with saline or CNO 30 min prior each testing. For testing, animals were placed in the apparatus, and the temperature was increased linearly from 45 to 55 °C at a rate of 5 °C/min. The trial was terminated when the animal jumped or after 2 min when the cutoff temperature of 55 °C was reached, whichever occurred first. The temperature, at which a hind paw low-intensity licking response or high-intensity jump response occurred, was determined. If no licking or jumping occurred, the parameter was set to its maximum value (55 °C).

**FMRI preparation and protocols.** FMRI measurements were conducted on a 4.7 T small animal MRT (Bruker Biospec, Bruker BioSpin MRI GmbH, Ettlingen, Germany). The MRT was equipped with a 200 mT/m gradient system, an actively decoupled RF-coil system for excitation and was operated by ParaVision software (V. 5.1, Bruker BioSpin MRI GmbH, Ettlingen, Germany). Animals were anesthetized for 4 min in 5% isoflurane in medical air and mounted on a specifically developed Plexiglass tray (Bruker BioSpin MRI GmbH, Ettlingen, Germany). An integrated water heating system helped to keep the body temperature constant at 37 °C. The head of the animal was fixed by the front teeth in a nose-mouth mask that prevented head movement and supplied low concentrations (0.7–1.5%) of isoflurane and oxygen/air for constant anesthesia during the whole imaging process. Anesthesia was adjusted during the measurement to keep the breathing rate between 90 and 120 breaths/min, known to allow best BOLD-contrast and minimal head movement.

To prevent exsiccation damage, the eyes of the animal were covered with an eye and nose ointment (Bepanthen, Bayer Vital GmbH, Leverkusen, Germany). For MR signal detection and good signal-to-noise ratio, a 3 cm 4-channel array head coil (Bruker BioSpin MRI GmbH, Ettlingen, Germany) with two holes was fitted to match the right hole with the optogenetic fiber implanted on the animal. The laser fiber was fixed to the implant afterward.

To verify the correct positioning of the animal inside the scanner, initial scout images, as well as typical adjustments to correct field inhomogeneities (shimming), were performed prior to the fMRI measurements. A fast gradient-echo echo planar imaging sequence (TR = 100 ms; $TE_{eff}$ = 25.3 ms; FOV = 15 mm × 15 mm; 1 slice; slice thickness 0.5 mm; matrix 64 × 64 voxels) was performed with 300 repetitions. Played as a video, it allowed detecting head movement of the animal. If so, the animal was remounted, and the positioning process restarted.

After the presets, one volume of 22 coronal slices (covering the brain from Bregma −2.06 mm to 1.42 mm) with an in-plane resolution of 0.234 mm × 0.234 mm was acquired using gradient-echo echo planar imaging (TR = 2000 ms; $TE_{ef}$ = 25.3 ms; FOV = 15 mm × 15 mm; slice thickness 0.5 mm; matrix 64 × 64 voxels) to reassure good image quality. If inevitable, the FastmapScout macro implemented in ParaVision 5.1 was used to correct local field inhomogeneities. Next, functional MRI measurements with 1950 volumes of these 22 slices were acquired during 65 min. Afterward, an anatomical dataset at identical slice positions was acquired in 16 min using T2-weighted rapid acquisition relaxation enhanced sequence (RARE; TR = 2000 ms; $TE_{eff}$ = 56 ms; k-space averaging 4; RAREFactor = 8; FOV = 15.1 mm × 15.1 mm; 22 slices; slice thickness 0.5 mm; matrix 256 × 256 voxel).

**OfMRI stimulation paradigm.** The right hind paw (ipsilateral to the implanted optogenetic fiber) of the animal was fixed with the dorsal side touching a computer-controlled Peltier heating device. Within 65 min, 16 heat stimuli of 50 °C were applied for 20 s each (5 s ramp, 15 s plateau) (Supplementary Fig. 2b). Every 2nd heat stimulus was combined with simultaneous laser application of 10 mW and 10 Hz frequency at a wavelength of 473 nm (termed co-stimulation). Every heat stimulus and combination were interspersed with one laser stimulus using the same settings. The stimulus interval was always 100 s.

**Wild-type heat stimulation paradigm.** Matching the ofMRI-experiment, the right hind paw of 29 wild-type mice (C57BL/6-background) were fixed to the computer-controlled Peltier heating device. Three sets of ascending innocuous (40 and 45 °C) and noxious (50 and 55 °C) temperatures were applied during a 50-min fMRI session. Beginning with a 2 min rest, each stimulus was applied for 20 s (5 s ramp, 15 s plateau), stimulus interval was 3 min 40 s.

**Data analysis and statistics**

*Behavioral data.* All behavioral data were obtained automatically from the aesthesiometer or the hot plate apparatus. For statistical analysis, two-way repeated-measures ANOVA (followed by Holm–Sidak post hoc tests) was used, as the injections were performed in the same animals, with the groups and injections (saline or CNO) as independent variables. The change in force or temperature was calculated based on the difference between CNO and saline for each individual

mouse. For this comparison, one-sample $t$ test was used with a theoretical mean zero. For between-group comparisons, one-way ANOVA (followed by Holm–Sidak post hoc tests) was used. All statistics were performed with GraphPad Prism version 8.

*Stimulus-driven (o)fMRI.* After the acquisition, the (o)fMRI datasets were resampled by averaging every two consecutive volumes to reduce noise, resulting in a $TR_{eff}$ of 4000 ms. This and most further analyses were performed using custom-programmed MagnAn (BioCom GbR, Uttenreuth in IDL, Harris Geospatial Solutions). Preprocessing, after discarding the first two volumes of the datasets avoiding MR saturation effects, comprised slice scan time correction (ascending interleaved, interpolation method cubic spline), motion correction to eliminate the minimal mouse head movement (registration to first brain volume; trilinear detection and sinc interpolation), spatial (Gaussian smoothing with a kernel size of 2 pixels) and temporal smoothing (linear and nonlinear high-pass filtering, kernel 12 s FWHM, FFT 9 cycles) was performed in Brainvoyager QX (Brain Innovation, Maastricht, Netherlands; V 2.8.2.2523). Correlation between the preprocessed signal time courses and the stimulation protocol was determined voxel-wise using the Two-Gamma HRF GLM algorithm of Brainvoyager, with separate predictors for each stimulus type. Further analysis was performed in MagnAn: to align the datasets for group analysis, all animals were registered to an anatomical reference via affine registration with 6 degrees of freedom (translation $x$-, $y$-, and $z$ axis, rotation $z$ axis, tilt in the $z$ axis, scale in $x$–$y$ axis).

The statistical parametric maps for each predictor obtained after GLM analysis were registered and corrected for multiple comparisons using FDR ($q = 0.05$) on the subject level. The resulting significantly activated voxels were labeled as belonging to one of 196 brain regions using a digital 3D modified Paxinos mouse brain atlas[31]. For each animal and each brain region, BOLD time courses of all significantly activated voxels were averaged. Next, event-related average time profiles of all eight presentations of the stimulus were calculated for the laser-heat co-stimulation, 10 timepoints before and 15 after the stimulation period (Fig. 2b and Supplementary Fig. 2b), resulting in one average time profile per brain region. Significant differences between the event-related averaged time courses of ChR2 groups and their respective GFP controls were calculated using a one-factor, repeated-measures ANOVA with subsequent Tukey HSD and Bonferroni correction for multiple comparisons. As an additional step, voxel-wise activation maps containing the maximum BOLD signal amplitude value during the whole experiment were calculated for each condition (Fig. 2a).

After removal of the global mean, the Pearson correlation coefficient $r$ was calculated between the full-length average time courses of all 196 brain regions for each animal. This yielded one correlation matrix per subject and predictor, representing the similarity of the time courses across all brain structures.

$R$-values had to be converted into normally distributed Fisher-$z$-values to calculate one mean correlation matrix per group and condition. The mean adjacency matrices were converted back into $r$-values, representing the functional connectivity[32] (FC) between the time courses (Supplementary Fig. 7).

Positive correlations (pCorrs) represent a high similarity of the time courses that reflect temporal synchrony in activation, and therefore represent a direct relation and interaction between brain regions. Negative correlations (nCorrs) are slightly less understood and implications are sometimes controversial. We agree with the assumption of Chen et al.[33] and Goelman et al.[34], that nCorrs reflect also regional co-activation, but delayed in time due to traveling longer distances. Therefore, nCorrs are found predominantly between cortical and non-cortical structures, and most of these structures are segregated by far from each other. We assume that they might also reflect possibly negative modulation indicating inhibitory effects here of the GABAergic nature of CEl PKCδ+ and SST+ neurons, or inhibition-disinhibition-effects between regions.

To allow for an optimal topological comparison, matrices had to be limited to contain the same number of connections. Across all stimulation conditions, 100 of the 196 identified brain structures showed an activation probability ≥50 percent. The adjacency matrices were thresholded therefore to contain the 500 strongest positive $r$-values (pCorrs) as well as the 500 lowest negative $r$-values (nCorrs) resulting in the frequently used $k$-value of 10 for the topological comparison (Supplementary Fig. 7, upper triangles).

Significant differences in FC matrices were calculated using a two-tailed Student's $t$ test, either homoscedastic or paired, as appropriate. Differences with $P \leq 0.05$ were accepted as significantly different.

*Comparison of heat processing of wild-type mice and PKCδ::ChR2/SST::ChR2.* In addition to the comparison between the ChR2 groups and the corresponding GFP controls, PKCδ::ChR2 and SST::ChR2 laser-heat co-stimulation were also compared to wild-type animals processing either innocuous 45 °C or noxious 50 °C. Preprocessing and graph-theoretical data evaluation of wild-type mice was done as described in the above section for both, nCorrs and pCorrs. Significant differences in FC matrices were calculated using a homoscedastic two-tailed Student's $t$ test. Differences with $P \leq 0.05$ were accepted as significantly different.

*Blinding.* The experimenter was not blind to surgery but blinded to the assignment of the behavioral groups. Behavioral scoring was done automatically by the apparatus software.

For fMRI, the analyst was not blinded, but the analysis was carried out with all animals taken together in formalized workflows to avoid bias effects. Following this workflow, no animal-specific input to introduce bias was possible.

Histological validation was performed not blinded to group assignment but blinded to the per animal outcome of behavioral or fMRI experiments.

**Reporting summary**. Further information on research design is available in the Nature Research Reporting Summary linked to this article.

## Data availability

Figure 1: sequencing data, published at GEO: GSE95154; Figs. 2 and 3: raw fMRI data are available upon request; Supplementary Fig. 1: sequencing data, published at GEO: GSE95154. Supplementary Fig. 3: raw histological data are available upon request; Supplementary Figs. 4, 5, and 7: raw fMRI data are available upon request; Supplementary Fig. 6: raw behavioral data is available upon request.

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

## Acknowledgements
We would like to thank the HistoPathology laboratory in Vienna Biocenter Core Facilities GmbH (VBCF) for histological services and Thomas Burkard and Maria Novatchkova from IMP/IMBA Bioinformatics Core Facility for analyzing NGS data. The VBCF preclinical Phenotyping Facility acknowledges funding from the Austrian Federal Ministry of Science, Research & Economy and the City of Vienna. A.H. was supported by the BMBF grants (NeuroImpa 01EC1403C and NeuroRad 02NUK034D). W.H. was supported by a grant from the European Community's Seventh Framework Programme (FP/2007-2013)/ERC grant agreement no. 311701, the Research Institute of Molecular Pathology (IMP), Boehringer Ingelheim, and the Austrian Research Promotion Agency (FFG).

## Author contributions
P.P., A.H., and W.H. designed the research; I.W., P.P., S.B., K.K., J.K., J.G., and A.H. performed the research; I.W., P.P., S.B., K.K., S.K., and A.H. analyzed the data; I.W., P.P., A.H., and W.H. wrote the paper.

## Competing interests
The authors declare no competing interests.
