## [Peer Review File · Communications Biology]

Reviewers' comments:

Reviewer #1 (Remarks to the Author):

The authors present an interesting examination of amygdala's modulation of acute pain perception, with demonstration of differentially recruited circuits by distinct cell types within lateral CeA (CeL). Overall, the findings suggest that CeL PKC+ expressing neurons modulate connectivity of 'bottom-up' networks to suppress nociception while SST+ expressing neurons in CeL engage 'top-down' networks to promote nociception. These findings build on previous papers dissecting the opposing roles of PKC+ vs. SST+ CeL neurons in pain perception, and add important new findings regarding brain-wide network dynamics. The authors apply appropriate experimental design, using reasonable analyses and controls, while also making suitable interpretations. Their results have implications for understanding amygdala's role in brain-wide pain modulation not previously thoroughly understood, and interestingly bridges whole brain analyses with cell-type specific targeting. As a result, this paper can have the potential to impact future causal circuit studies in pain perception. Though enthusiasm is generally high, some concerns with regard to interpretation of results and clarification of analyses should be addressed:

1. The deep sequencing results from FACS-sorted PKC+ and SST+ expressing CeL neurons are compelling and add interesting support to overall conclusions, but the discussion falls short of tying them in with the other results. The manuscript would benefit from including more discussion on differentially expressed nociception-related genes, and how their divergent mechanisms potentially lead to the observed anti vs. pro-nociception by the different cell types. Relatedly, on page 3, lines 61-64 mentions that these cell types differentially express genes related to 'central transmission of pain' vs. 'peripheral modulation'. Please expand more on what is meant by central vs. peripheral modulation here.
2. Laser plus heat co-stimulation induces an interesting BOLD amplitude increase in sensory cortex activity among both groups, but appears to be at a different time course between PKC:cre and SST:cre groups (Fig 2b), with PKC:cre mice displaying a delayed BOLD amplitude in sensory cortex beyond laser stimulation. Is this true? And if so, it might be worth discussing briefly, and perhaps expanding on implications (i.e., discussing disinhibition via bottom up mechanisms).
3. In Figure 1b, I would suggest also adding a legend depicting how each shade of red/blue corresponds to the particular DElog2 fold change (depicted in Extended fig 1), going from lighter shades of red or blue to darker shades of red or blue.
4. Please describe why the particular laser stimulation parameters (10Hz, 10mW, 20s duration?) were chosen, and any relevance to their use in behavioral pain assays.
5. Over the course of the 16 heat trials throughout the 65min session, was there a habituation effect observed in BOLD response amplitude? Based on analyses, it is unclear whether all trials were included and averaged, or if you were able to pull out an order effect? This would potentially be interesting with regard to networks engaged in first heat exposure vs. last/multiple exposures, for example.
6. More discussion that includes known differential intra-CeA mechanisms of SST vs PKC neurons (i.e., disinhibition by PKC+ neurons of CeM output neurons) that lead to differential bottom-up vs top down processes observed between the two groups is needed.
7. Although the methods section describes timepoints used for BOLD amplitudes analyses, it would also be helpful if timepoints were clearly defined in the x axis on Figure 2b.
8. The authors state on page 6 line 135 that "CeL PKC stimulation at 52 degrees results in FC pattern that overlap with those of wild type animals stimulated with innocuous 45 degrees." Please unpack/expand on this statement, and clarify how the authors came to this conclusion.
9. Please define FC units where they appear in Figures 3a and Extended data Figure 5b.
10. Could the authors discuss possible reasons for CNO-induced enhancement in temperature tolerance in the hot plate test among GFP controls? How does that effect interpretation of CNO effects in PKC group?

Reviewer #3 (Remarks to the Author):

In this impressive piece of work the authors report a thorough in-depth investigation of the role of

the central amygdala circuitry in nociceptive processing. By expression analysis of pain-related genes, the authors identified protein kinase C delta and somatostatin expressing neurons as important populations. Using a combination of optogenetic fMRI and DREADD chemogenetic activation combined with behavioral experiments, the authors have identified subnetworks by which these populations modulate brain dynamics via hierarchical top-down and bottom-up mechanisms, such as reduced antinociceptive modulation of nociceptive Bs signals, or uncoupling of aversive behavioral states from thalamocortical nociceptive representation.

The study is adequately designed, methods are state of the art, group sizes of animals are appropriate, and statistical analysis is valid.

The major strength of the paper is that it provides a view of the role of central amygdala in nociceptive processing with unprecedented depth.

I have no major concerns, and recommend publication after the authors have addressed a few minor points.

Readability of the paper could be improved in several points:

1. On the first five pages, the authors do not mention that also a chemogenetic approach was used in addition to the ofMRI approach. This is confusing to the reader who takes a glimpse at the supplements or methods before finishing the main part.

Similarly in the methods. The M3-group is mentioned before the authors explained what M3 is or that they used a chemogenetic approach.

I suggest adding a sentence introducing the chemogenetic approach in line 76. Further, the description should be added to the methods in line 499.

2. Another potentially confusing point is the use of the term laser stimulation. It would be better to speak of optogenetic stimulation, since laser stimulation might as well be applied as heat stimulus to the paw. Strictly speaking laser application/illumination would be correct, since in the GFP controls, no defined stimulation is intended.

3. A number of cryptic names for routines, algorithms or variables are used without introducing them, e.g. in legend to Fig. 1, or between lines 426 and 434.

4. In legend to Fig. 2, the authors briefly discuss potential artefacts from heat or light related signal changes, and state that those are expected to be consistent across groups. Recent studies have shown that this is not necessarily the case, and a simple way to identify those signal changes is by their typical time course. The authors should follow this procedure and state that signal changes in regions close to laser illumination did not show such typical time courses – in case the authors have not done so, they should verify and state accordingly.

5. Heat stimulation experiments of wt mice is not described in the methods. Further, in line 134 a temperature of 52°C is stated and in line 494 50°C.

6. In line 169 the authors state Highly translational fMRI FC analysis. What makes this procedure particularly translational?

7. Line 279 should read right column.

8. In line 482, TR is given twice.

9. In Ext. Fig. 3 a and b, axes are labeled with virus expressing cells. Obviously, not the virus is expressed, but the opsin/GFP.

10. Second line in legend to Ext. Fig. 4 should read ...compared to their...

11. Legend to Suppl. Table 2 should state ...behavior combined with DREADD chemogenetic activation.

Response to the Reviews of the Manuscript “Central amygdala circuitry modulates nociceptive processing through differential hierarchical interaction with affective network dynamics”

Vienna, April 16, 2021

To the Reviewers:

We are grateful for the very thoughtful reviews and the overall positive reception of the manuscript by the Reviewers, and for recognizing the optogenetic fMRI experiments and mesoscale mechanisms as timely contribution to the field.

We took all notes into account and are positive that this constructive discussion improved the manuscript in many ways.

Best wishes,

Dr. Wulf Haubensak

Reviewer #1 (Remarks to the Author):

The authors present an interesting examination of amygdala's modulation of acute pain perception, with demonstration of differentially recruited circuits by distinct cell types within lateral CeA (CeL). Overall, the findings suggest that CeL PKC+ expressing neurons modulate connectivity of 'bottom-up' networks to suppress nociception while SST+ expressing neurons in CeL engage 'top-down' networks to promote nociception. These findings build on previous papers dissecting the opposing roles of PKC+ vs. SST+ CeL neurons in pain perception, and add important new findings regarding brain-wide network dynamics. The authors apply appropriate experimental design, using reasonable analyses and controls, while also making suitable interpretations. Their results have implications for understanding amygdala's role in brain-wide pain modulation not

previously thoroughly understood, and interestingly bridges whole brain analyses with cell-type specific targeting. As a result, this paper can have the potential to impact future causal circuit studies in pain perception. Though enthusiasm is generally high, some concerns with regard to interpretation of results and clarification of analyses should be addressed:

1. The deep sequencing results from FACS-sorted PKC+ and SST+ expressing CeL neurons are compelling and add interesting support to overall conclusions, but the discussion falls short of tying them in with the other results. The manuscript would benefit from including more discussion on differentially expressed nociception-related genes, and how their divergent mechanisms potentially lead to the observed anti vs. pro-nociception by the different cell types. We thank this Reviewer for this excellent suggestion.

We amended an additional analyses lines 70-73 and Supplementary Fig. 1b to highlight the molecular distribution of the CE cell types on nociception and how this could support (a differential) impact of these two classes related to pain. We have discussed a possible scenario of how this molecular makeup biases the relay of CE afferent nociceptive signals in lines 178-186.

Relatedly, on page 3, lines 61-64 mentions that these cell types differentially express genes related to 'central transmission of pain' vs. 'peripheral modulation'. Please expand more on what is meant by central vs. peripheral modulation here. We thank this Reviewer for highlighting a potential misunderstanding/unclear terminology.

We have rephrased central vs. peripheral in the text: the terms were referring to node network position and not central vs. peripheral processing of nociception/pain.

2. Laser plus heat co-stimulation induces an interesting BOLD amplitude increase in sensory cortex activity among both groups, but appears to be at a different time course between PKC:cre and SST:cre groups (Fig 2b), with PKC:cre mice displaying a delayed BOLD amplitude in sensory cortex beyond laser stimulation. Is this true? And if so, it might be worth discussing briefly, and perhaps expanding on implications (i.e., discussing disinhibition via bottom up mechanisms).

Thank you very much for pointing this out. The seemingly delayed BOLD amplitude in the PKC-groups is indeed an artifact due to the calculation of differences between ChR2- and GFP -groups. As shown below in green, the time courses of PKC::GFP and PKC::ChR2 are highly similar in rise and amplitude, but the decay is slightly different.

As we've calculated the difference between Chr2- and GFP -groups, this apparently delayed amplitude survives. We are sorry about this confusion.

To make it clearer, we've adjusted the label of the y-axis to 'Difference in BOLD-Signal [%]'. If the reviewers prefer so, we could also edit Fig. 2b to show the single group amplitudes instead of the differences, but then due to spatial limitations, as a figure of its own instead as a panel of Fig. 2. Shown below are the average time profiles for sensory cortex of (left in greens) PKC::GFP/PKC::ChR2 and (right in reds) SST::GFP/SST::ChR2. Below is the corresponding difference between the Chr2-group and the GFP-group. The error bars display the standard-error of the mean across animals and brain regions.

3. In Figure 1b, I would suggest also adding a legend depicting how each shade of red/blue corresponds to the particular DElog2 fold change (depicted in Extended fig 1), going from lighter shades of red or blue to darker shades of red or blue.

This is an excellent suggestion.

We have modified the presentation accordingly.

4. Please describe why the particular laser stimulation parameters (10Hz, 10mW, 20s duration?) were chosen, and any relevance to their use in behavioral pain assays.

This is an excellent point. We used frequency and duration that were most effective on when interfering with CE function in fMRI pilot experiments (data not

shown). Moreover, typically, 10 Hz correspond to the max firing rate of the neurons ¹ *in-vivo*, 10 mW laser power is typically used in behavioral experiments. This intensity is below (70 mW/mm² for 20 ms pulse duration and 20 % duty cycle) the threshold for tissue heating (156 mW/mm² for 100 % duty cycle) at our 20 % duty cycles (Reviewer #3 Major point).

We note that the pain assays were performed with chemogenetic perturbations for practical reasons (easier to handle in these assays in our hands). However, we do believe that both optogenetic and chemogenetic methods used result in net activation of the target population, as both of these manipulations have been consistently and interchangeably used by us and others, to manipulate CE circuitry (e.g. ²⁻⁵).

5. Over the course of the 16 heat trials throughout the 65min session, was there a habituation effect observed in BOLD response amplitude? Based on analyses, it is unclear whether all trials were included and averaged, or if you were able to pull out an order effect? This would potentially be interesting with regard to networks engaged in first heat exposure vs. last/multiple exposures, for example. We apologize for being unclear here in the methods section.

We have edited the corresponding paragraph in order to be more precise. For calculation of the average time profiles, signals of all eight presentations of the laser + heat co-stimulation were averaged. We've also looked at the single repetition amplitudes of representative regions (shown below for the laser + heat co-stimulation) and found no time-dependent modulatory effects. Data below shows average BOLD-signal amplitude per stimulus presentation across the animals + standard-error of the mean.

6. More discussion that includes known differential intra-CeA mechanisms of SST vs PKC neurons (i.e., disinhibition by PKC+ neurons of CeM output neurons) that lead to differential bottom-up vs top down processes observed between the two groups is needed.

We thank the Reviewer for this excellent suggestion.

We have amended the discussion lines 206-218 in this regard.

7. Although the methods section describes timepoints used for BOLD amplitudes analyses, it would also be helpful if timepoints were clearly defined in the x axis on Figure 2b.

This is an excellent suggestion.

We have added the 'timepoint'-label to each x-axis, instead of just the lowest one. This should render the figure easier to comprehend.

8. The authors state on page 6 line 135 that "CeL PKC stimulation at 52 degrees results in FC pattern that overlap with those of wild type animals stimulated with innocuous 45 degrees." Please unpack/expand on this statement, and clarify how the authors came to this conclusion.

Thank you very much for this comment. First, we'd like to apologize for making a mistake, heat stimulation of wt animals was 45 °C and 50 °C, not 52 °C as stated.

We now added an additional analysis (see lines 139-148), and compared wt FC under stimulation with innocuous 45 °C and noxious 50 °C respectively, directly to PKC δ^+ /SST $^+$ laser + heat 50 °C co-stimulation. Accordingly, we changed Supplementary Fig. 5 to contain the new data. Results showed, that PKC δ^+ co-stimulation showed stronger FC (positive values in Supplementary Fig. 5) compared to wt 45 °C, but weaker FC (negative values in Supplementary Fig.) than wt 50 °C. SST $^+$ co-stimulation on the other hand, displayed way stronger FC than wt 45 °C, and even stronger FC than wt 50 °C (here only in hippocampal connections due to limiting the data to contain at least 4 connections). This led us to the conclusion, that PKC δ^+ co-stimulation reduced perception of noxious heat to a level between wt 45 and 50 °C. This effect could not be shown for SST $^+$ co-stimulation.

9. Please define FC units where they appear in Figures 3a and Extended data Figure 5b.

Thank you for pointing this out.

The legends are now revised accordingly.

10. Could the authors discuss possible reasons for CNO-induced enhancement in temperature tolerance in the hot plate test among GFP controls? How does that effect interpretation of CNO effects in PKC group?

We have no good explanation for this effect, as the experiments were not performed in a balanced manner. It could be behavioral adaptation and/or CNO

effects. However, the trend describes a between group effect under similar treatment. We have amended the results section in lines 156-163 in this regard.

If this reviewer agrees, we feel that we can omit the results from the manuscript.

Reviewer #3 (Remarks to the Author):

In this impressive piece of work the authors report a thorough in-depth investigation of the role of the central amygdala circuitry in nociceptive processing. By expression analysis of pain-related genes, the authors identified protein kinase C delta and somatostatin expressing neurons as important populations. Using a combination of optogenetic fMRI and DREADD chemogenetic activation combined with behavioral experiments, the authors have identified subnetworks by which these populations modulate brain dynamics via hierarchical top-down and bottom-up mechanisms, such as reduced antinociceptive modulation of nociceptive Bs signals, or uncoupling of aversive behavioral states from thalamocortical nociceptive representation. The study is adequately designed, methods are state of the art, group sizes of animals are appropriate, and statistical analysis is valid. The major strength of the paper is that it provides a view of the role of central amygdala in nociceptive processing with unprecedented depth. I have no major concerns, and recommend publication after the authors have addressed a few minor points. Readability of the paper could be improved in several points:

1. On the first five pages, the authors do not mention that also a chemogenetic approach was used in addition to the ofMRI approach. This is confusing to the reader who takes a glimpse at the supplements or methods before finishing the main part. Similarly in the methods. The M3-group is mentioned before the authors explained what M3 is or that they used a chemogenetic approach. I suggest adding a sentence introducing the chemogenetic approach in line 76. Further, the description should be added to the methods in line 499.

Thank you very much for this comment.

We have added the chemogenetic approach to the introduction (lines 50-52) as well as to the methods section (lines 416-420).

2. Another potentially confusing point is the use of the term laser stimulation. It would be better to speak of optogenetic stimulation, since laser stimulation might as well be applied as heat stimulus to the paw. Strictly speaking laser application/illumination would be correct, since in the GFP controls, no defined stimulation is intended.

Thank you for indicating that the term 'laser stimulation' might be confused with noxious laser heat stimulation. That was unintended. We switched to 'laser application' wherever feasible.

3. A number of cryptic names for routines, algorithms or variables are used without introducing them, e.g. in legend to Fig. 1, or between lines 426 and 434. This is a very good suggestion.

We have amended and revised this section. Note that many of these terms refer to standard algorithms and software packages⁶.

4. In legend to Fig. 2, the authors briefly discuss potential artefacts from heat or light related signal changes, and state that those are expected to be consistent across groups. Recent studies have shown that this is not necessarily the case, and a simple way to identify those signal changes is by their typical time course. The authors should follow this procedure and state that signal changes in regions close to laser illumination did not show such typical time courses - in case the authors have not done so, they should verify and state accordingly. Thank you very much for this very valuable note.

We re-checked the average time courses of the right amygdala for the typical heat-induced logarithmic time courses⁷ in all four experimental groups, but found only the hemodynamic response-like signal phenotype shown below. We have added this finding to the respective figure legend of Fig. 2 for clearer explanation. Data below shows average BOLD-signal amplitude across the animals + standard-error of the mean.

5. Heat stimulation experiments of wt mice is not described in the methods. Further, in line 134 a temperature of 52°C is stated and in line 494 50°C. Thank you very much for pointing this out.

We have added a section in the methods in lines 567-572. We apologize for the confusion about the stimulation temperature, this was a mistake (see comment 8 to Reviewer #1). Stimulation temperature was in all cases 50 °C, not 52 °C.

6. In line 169 the authors state highly translational fMRI FC analysis. What makes this procedure particularly translational? fMRI is fully human compatible, as it is un-invasive, does not require the use of external contrast agents or is harmful in any other known way. The analysis of acquired fMRI data, especially novel graph-theoretical analyses, can also be applied to functional human data, even post-hoc after the initial measurement was done any time ago, as it does not require special presets or conditions. Conclusively, the exact same analysis workflow can be used for animal and human data, which make the results highly comparable^{8,9}.

7. Line 279 should read right column.
Thank you.

The phrase is now edited from “row” to “column”.

8. In line 482, TR is given twice.
Thank you.

The repetition is now deleted.

9. In Ext. Fig. 3 a and b, axes are labeled with virus expressing cells. Obviously, not the virus is expressed, but the opsin/GFP.
Thank you.

The labels have been corrected.

10. Second line in legend to Ext. Fig. 4 should read ...compared to their...
Thank you.

Thank you, the phrase was corrected.

11. Legend to Suppl. Table 2 should state ...behavior combined with DREADD chemogenetic activation.

Thank you.

We added the statement accordingly.

Literature:

1. Ciochi S, Herry C, Grenier F, et al. Encoding of conditioned fear in central amygdala inhibitory circuits. *Nature*. 2010;468(7321):277-282. doi:10.1038/nature09559
2. Griessner J, Pasięka M, Böhm V, et al. Central amygdala circuit dynamics underlying the benzodiazepine anxiolytic effect. *Mol Psychiatry*. 2021;26(2):534-544. doi:10.1038/s41380-018-0310-3
3. Kargl D, Kaczanowska J, Ulonska S, et al. The amygdala instructs insular feedback for affective learning. *eLife* 2020;9:e60336. 2020;9:1-36. doi:10.7554/eLife.60336
4. Yu K, Ahrens S, Zhang X, et al. The central amygdala controls learning in the lateral amygdala. *Nat Neurosci*. 2017;20(12):1680-1685. doi:10.1038/s41593-017-0009-9
5. Tye KM, Prakash R, Kim SY, et al. Amygdala circuitry mediating reversible and bidirectional control of anxiety. *Nature*. 2011;471(7338):358-362. doi:10.1038/nature09820
6. Contreras-López O, Moyano TC, Soto DC, Gutiérrez RA. Step-by-step construction of gene co-expression networks from high-throughput Arabidopsis RNA sequencing data. In: *Methods in Molecular Biology*. ; 2018. doi:10.1007/978-1-4939-7747-5_21
7. Albers F, Wachsmuth L, Schaché D, Lambers H, Faber C. Functional mri readouts from bold and diffusion measurements differentially respond to optogenetic activation and tissue heating. *Front Neurosci*. Published online 2019. doi:10.3389/fnins.2019.01104
8. Neely GG, Hess A, Costigan M, et al. A Genome-wide Drosophila screen for heat nociception identifies $\alpha 2\delta 3$ as an evolutionarily conserved pain gene. *Cell*. Published online 2010. doi:10.1016/j.cell.2010.09.047
9. Hess A, Axmann R, Rech J, et al. Blockade of TNF- α rapidly inhibits pain responses in the central nervous system. *Proc Natl Acad Sci*. 2011;108(9):3731 LP - 3736. doi:10.1073/pnas.1011774108

REVIEWERS' COMMENTS:

Reviewer #1 (Remarks to the Author):

Overall, the authors did an excellent job alleviating my initial concerns regarding analyses and interpretation of results. Their revised manuscript is greatly improved, and I feel makes an important and timely contribution to the field.

Regarding the authors' response to point #2, and whether or not to add the figure showing single-group BOLD amplitudes to figure 2 in addition to/or instead of differences from GFP, I don't feel there is any need for that. The addition they added in the text of the figure legend for Figure 2 describing the delay in decay of BOLD amplitude in sensory cortex of the PKCdelta:Cre group is sufficient.

Regarding the authors' response to point #10 on whether or not to keep the results of DREADD activation on heat response thresholds, and specifically for the GFP group, I feel that the results should remain. The explanation authors provide for potential reasons for CNO effect among GFP controls (including habituation) are sufficient, and the analyses showing a difference between groups with CNO application are also sufficient and important.

Reviewer #3 (Remarks to the Author):

The authors have responded to all points and resolved all concerns I had about the previous version. I am happy to recommend publication in the present form.